

# Comparative proteomic analysis of pepper (*Capsicum annuum* L.) seedlings under selenium stress

Chenghao Zhang[1,2], Baoyu Xu[1], Wei Geng[3], Yunde Shen[4], Dongji Xuan[4], Qixian Lai[5], Chenjia Shen[6], Chengwu Jin[7] and Chenliang Yu[1]

[1] Institute of Agricultural Equipment, Zhejiang Academy of Agricultural Sciences, Hangzhou, Zhejiang, China
[2] Key Labortatory of Creative Agricultrue, Ministry of Agriculture, Zhejiang Academy of Agricultural Science, Hangzhou, Zhejiang, China
[3] Vegetable Research Institute, Zhejiang Academy of Agricultural Sciences, Hangzhou, Zhejiang, China
[4] College of Mechanical and Electrical Engineering, Wenzhou University, Wenzhou, Zhejiang, China
[5] Key Labortatory of Creative Agricultrue, Ministry of Agriculture, Zhejiang Academy of Agricultural Sciences, Hangzhou, Zhejiang, China
[6] College of Life and Environmental Science, Hangzhou Normal University, Hangzhou, Zhejiang, China
[7] School of Food Engineering, Ludong University, Yantai, Shandong, China

Corresponding authors
Chengwu Jin, jinchwu@ldu.edu.cn
Chenliang Yu, 21007030@zju.edu.cn

## ABSTRACT

Selenium (Se) is an essential trace element for human and animal health. Se fertilizer has been used to increase the Se content in crops to meet the Se requirements in humans and animals. To address the challenge of Se poisoning in plants, the mechanisms underlying Se-induced stress in plants must be understood. Here, to elucidate the effects of Se stress on the protein levels in pepper, we used an integrated approach involving tandem mass tag labeling, high performance liquid chromatography fractionation, and mass spectrometry-based analysis. A total of 4,693 proteins were identified, 3,938 of which yielded quantitative information. Among them, the expression of 172 proteins was up-regulated, and the expression of 28 proteins was down-regulated in the Se/mock treatment comparison. According to the above data, we performed a systematic bioinformatics analysis of all identified proteins and differentially expressed proteins (DEPs). The DEPs were most strongly associated with the terms "metabolic process," "posttranslational modification, protein turnover, chaperones," and "protein processing in endoplasmic reticulum" according to Gene Ontology, eukaryotic orthologous groups classification, and Kyoto Encyclopedia of Genes and Genomes enrichment analysis, respectively. Furthermore, several heat shock proteins were identified as DEPs. These results provide insights that may facilitate further studies on the pepper proteome expressed downstream of the Se stress response. Our data revealed that the responses of pepper to Se stress involve various pathways.

## INTRODUCTION

Selenium (Se) is a trace element that is essential for human and animal health, and it is an active component of numerous enzymes in human metabolism (*Sager, 2006*; *Semnani et al., 2010*). Because of its important protective effects in animals and plants, many studies on Se have been conducted in a broad range of fields including medicine, agriculture, and nutrition (*Chen et al., 2002*; *Thavarajah, Ruszkowski & Vandenberg, 2008*). Although there is no direct evidence that Se is necessary for plant growth, Se plays a key role in plant growth and development. Se can improve antioxidant enzyme activity and enhance the tolerance of *Rumex patientia* × *R. tianshanicus* seedlings to salt stress (*Kong, Wang & Bi, 2005*). In addition, Se enhances plant resistance to abiotic stresses, including heavy metals (*Kumar et al., 2012*), waterlogging (*Wang, 2011*), chilling (*Chu, Yao & Zhang, 2010*), high temperature (*Djanaguiraman, Prasad & Seppanen, 2010*) and drying (*Pukacka, Ratajczak & Kalemba, 2011*). Se also plays a critical role in plant resistance to biological stress. Plants with high Se content in grasslands can resist invasion by herbivores (*Quinn et al., 2008*).

The properties of Se facilitate the formation of stable compound structures with multiple oxidation states (+2, +4, and +6), covalent bonding to non-metals (such as carbon), and strong coordination with metals such as cadmium (*Fernandes et al., 2018*). The main forms of Se taken up by plants are selenate (VI) and selenite (IV), whereas the Se forms in soil are influenced by the soil pH and oxidation potential (*Elrashidi et al., 1987*). The uptake and transport mechanisms of the two major valences of Se ($SeO_4^{2-}$ and $SeO_3^{2-}$) in soil differ (*White, 2016*). The chemical properties of selenate and sulfate are similar (*Shibagaki et al., 2002*). These compounds are antagonistic during plant uptake, and the sulfate transporter regulates the uptake of selenite (*Shibagaki et al., 2002*; *El Kassis et al., 2007*; *White, 2016*). Currently, the mechanism of selenite absorption by plants is unclear. Most studies suggest that the mechanism of absorption of selenite is similar to that of phosphate, but selenite absorption is negatively correlated with phosphate absorption (*Zhang et al., 2014*; *Song et al., 2017*).

At present, Se poisoning incidents in plants have rarely been reported. Preliminary studies have shown that the toxic effect of Se on plants is similar to that of heavy metals to some extent and can hinder plant growth and metabolism. In agricultural environments, excessive Se has been found to decrease radish seeds by 14% and radish yield by 8–9% (*Hladun et al., 2013*). Se stress in barley hinders plant growth and significantly decreases fresh weight, water content, and photosynthetic capacity (*Molnárová & Fargašová, 2009*). *Paciolla, De Leonardis & Dipierro (2011)* have observed that Se treatment (8–16 mg/L) significantly inhibits barley germination (*Paciolla, De Leonardis & Dipierro, 2011*). In addition, Se (four to six mg/L) significantly inhibits root and bud growth in soybean seedlings, whereas root growth in lettuce and ryegrass is completely inhibited even at a concentration of one mg/L (*Hartikainen et al., 1997*; *Aggarwal et al., 2011*). The above

results demonstrate that Se causes toxicity in plants. However, current understating of the molecular mechanism of Se toxicity remains limited (*El-Ramady et al., 2015*; *Galinha et al., 2015*; *Jia et al., 2019*). Elemental Se and Se compounds are increasingly accumulating in surface soil and water. Excessive amounts of Se pose a potential risk in agricultural production (*Kuppusamy et al., 2018*; *Jia et al., 2019*).

Pepper (*Capsicum annuum* L.), an economically important vegetable in the *Solanaceae* family, has been used as a spice in China and Korea for decades (*Choi et al., 2005*). Recently, a novel tandem mass spectrometry (MS/MS)-based tandem mass tag (TMT) labeling strategy was developed for large-scale protein quantification (*Hao et al., 2017*; *Xu et al., 2017*). Relatively limited proteomic data on pepper under Se stress have been reported. In the present study, we used a TMT labeling-based quantitative proteomics approach to identify differentially expressed proteins (DEPs) under Se treatment. Our data enabled the identification and exploration of the roles of candidate proteins associated with Se stress resistance.

## MATERIALS AND METHODS

### Plant materials and Se treatments

Pepper seeds (*C. annuum* 8 #, a cultivar provided by pepper breeding group in Fujian Agriculture and Forestry University) were sterilized with 1% sodium hypochlorite for 30 min and grown in steam-sterilized soil. The seedlings were grown in a greenhouse under the following conditions: 12 h light (150 $\mu m^2 \ s^{-1}$) at 26 °C, 12 h dark at 23 °C, and relative humidity of 60%. Seedlings were irrigated with half-strength Hoagland solution (pH 5.6). Pepper plants at the four true leaf stages were used for Se treatment. Seedlings were sprayed with half-strength Hoagland solution containing 0 or 100 ppm $Na_2SeO_4$. After 24 h, the shoots were collected for protein extraction.

### Protein extraction and trypsin digestion

Samples were removed from storage at −80 °C, and fixed amounts of tissue samples were ground to powder while liquid nitrogen was added. The samples in each group were treated with four volumes of phenol extraction buffer (containing 10 mM dithiothreitol, 1% protease inhibitor, and two mM EDTA), then sonicated three times on ice with a high intensity ultrasonic processor (Scientz, Ningbo, China). The supernatant was centrifuged for 10 min at 4 °C and 5,500×*g* with an equal volume of Tris equilibrium phenol. The supernatant was collected and precipitated overnight with five volumes of 0.1M ammonium acetate/methanol. The protein precipitate was washed with methanol and acetone successively. Finally, the precipitate was re-dissolved in 8M urea, and the protein concentration was determined with a BCA kit (code P0010; Beyotime, Beijing, China) according to the manufacturer's instructions.

For digestion, the final concentration of dithiothreitol in protein solution was five mM, and reduction was performed by incubation at 56 °C for 30 min. The mixture was then alkylated with 11 mM iodoacetamide at room temperature for 15 min. Finally, the urea concentration of the sample was diluted to less than 2M by addition of 100 mM triethyl ammonium bicarbonate. Trypsin was added at a mass ratio of 1:50 (trypsin:protein), and

enzymatic hydrolysis was carried out overnight at 37 °C. Trypsin was then added at a mass ratio of 1:100 (trypsin:protein), and enzymatic hydrolysis continued for 4 h.

## TMT labeling and HPLC fractionation

The trypsinized peptide segments were desalted with a Strata X C18 column (Phenomenex, Torrance, CA, USA) and then freeze-dried under vacuum. The peptides were dissolved in 0.5M triethyl ammonium bicarbonate and labeled with a TMT kit (ThermoFisher, Shanghai, China) according to the manufacturer's instructions. The procedure was as follows: the labeled reagent was dissolved in acetonitrile after thawing, incubated at room temperature for 2 h after mixing with the peptide segments, then desalinated after mixing with the labeled peptide segment and freeze-dried in a vacuum.

Peptide segments were classified with high pH reverse-phase high performance liquid chromatography (HPLC) with an Agilent 300 Extend C18 column (five μm diameter, 4.6 mm inner diameter, 250 mm length) (Agilent, Shanghai, China). The gradient of peptide segments was 8–32% acetonitrile (pH 9.0), and more than 60 min was required to separate the peptide segments into 60 components. The peptide segments were then merged into 18 components, which were then vacuum freeze-dried.

## LC-MS/MS analysis

The tryptic peptides were dissolved in 0.1% formic acid (solvent A) and directly loaded onto an Acclaim PepMap 100 reverse-phase pre-column (ThermoFisher, Shanghai, China). The gradient comprised an increase from 6% to 23% solvent B (0.1% formic acid in 98% acetonitrile) over 26 min, from 23% to 35% in 8 min, and to 80% in 3 min; then the concentration was held at 80% for the last 3 min. All steps were performed at a constant flow rate of 500 nL/min on an EASY-nLC 1000 ultra-HPLC system.

After equilibration, the peptides were ionized with an NSI ion source and analyzed by MS/MS (Q ExactiveTM mass spectrometer; ThermoFisher, Shanghai, China) coupled online to ultra-HPLC. The ion source voltage was set to 2.0 kV, and the parent ions of peptide segments and their secondary fragments were detected and analyzed with a high resolution Orbitrap. The scanning range of the primary MS was set to 350–1,800 m/z, and the scanning resolution was set to 70,000. The scanning range of the secondary MS was set to 100 m/z, and the scanning resolution of the secondary MS was set to 17,500. The data acquisition mode used a data-dependent scanning program; that is, the first 20 peptide ions with the highest signal intensity were selected to enter the higher collision dissociation collision pool in turn after the first scan, and the fragmentation energy was 28% for fragmentation. Similarly, secondary MS analysis was carried out sequentially. To improve the efficiency of MS, AGC was set to 5E4, the signal threshold was set to 2E4, the maximum injection time was set to 100 ms, and the dynamic elimination time of MS/MS scanning was set to 30 s to avoid repeated scanning of parent ions.

## Database search

The resulting MS/MS data were processed and searched against a public proteome *C. annuum* database (https://www.uniprot.org/uniprot/?query=proteome:UP000222542)

with the Maxquant search engine (v.1.5.2.8, https://maxquant.org/) concatenated with a reverse decoy database. The retrieval parameter settings were as follows: trypsin/P was used for digestion; the number of missing digestion sites was set to two; and the minimum length of peptide segments was set to seven amino acids. The maximum number of modifications of the peptide segment was set to five. The mass error tolerance of the primary parent ions of the first search and main search was set to 20 and 5 ppm, respectively. The mass error tolerance was 0.02 Da. Cysteine alkylation was set as a fixed modification. Oxidation on Met was specified as a variable modification. The quantitative method was set to TMT-6plex, and the false discovery rate for protein identification and PSM identification was set to 1%.

## Protein annotation

The Gene Ontology (GO) annotation proteome was derived from the UniProt-GOA database (http://www.ebi.ac.uk/GOA/). Subsequently, proteins were classified according to GO annotation on the basis of three categories: biological process, cellular component, and molecular function. The Kyoto Encyclopedia of Genes and Genomes (KEGG) database was used to annotate protein pathways. KEGG online service tools KAAS (http://www.genome.jp/kaas-bin/kaas_main) were used to annotate the KEGG database descriptions of proteins, and then the annotation results were mapped on the KEGG pathway database by using the KEGG online service tool KEGG mapper (http://www.kegg.jp/kegg/mapper.html). For domain annotation, the InterProScan database (http://www.ebi.ac.uk/interpro/) was used to annotate the domain functional descriptions of identified proteins.
For subcellular localization, WoLFSPORT (a subcellular localization predication software (http://www.genscript.com/psort/wolf_psort.html) was used. WoLFSPORT is an updated version of PSORT/PSORT II for the prediction of eukaryotic sequences.

## Statistical analysis

For GO enrichment and pathway analysis, a two-tailed Fisher's exact test was used to test the enrichment of the DEPs against all identified proteins. A GO term or KEGG pathway with a $P$-value < 0.05 was considered significant. The proteins with TMT intensity values were considered quantified, and the minimal PIF was set as 0.75. Statistical analyses were carried out in SPSS ver. 19.0 (SPSS Inc., Chicago, IL, USA). All reported values represent the averages of three replicates with the standard deviation (mean ± SD).

## Quantitative real-time PCR validation

Total RNA was extracted using a Ultrapure RNA Kit according to the manufacturer's protocol (Code: CW0597, CWBIO, Beijing, China). First-strand cDNA synthesis was carried out using a SuperScript™ IV First-Strand Synthesis System according to the manufacturer's protocol (Code:18091050; ThermoFisher, Waltham, MA, USA). QRT-PCR was performed on ABI 7500 Real-Time PCR System (Roche, Basel, Switzerland) using UltraSYBR Mixture(High ROX)Kit (Code: CW2602, CWBIO, Beijing, China) with the primers listed in Table S1. The CaACTIN (Capana12g001934) was used as an internal

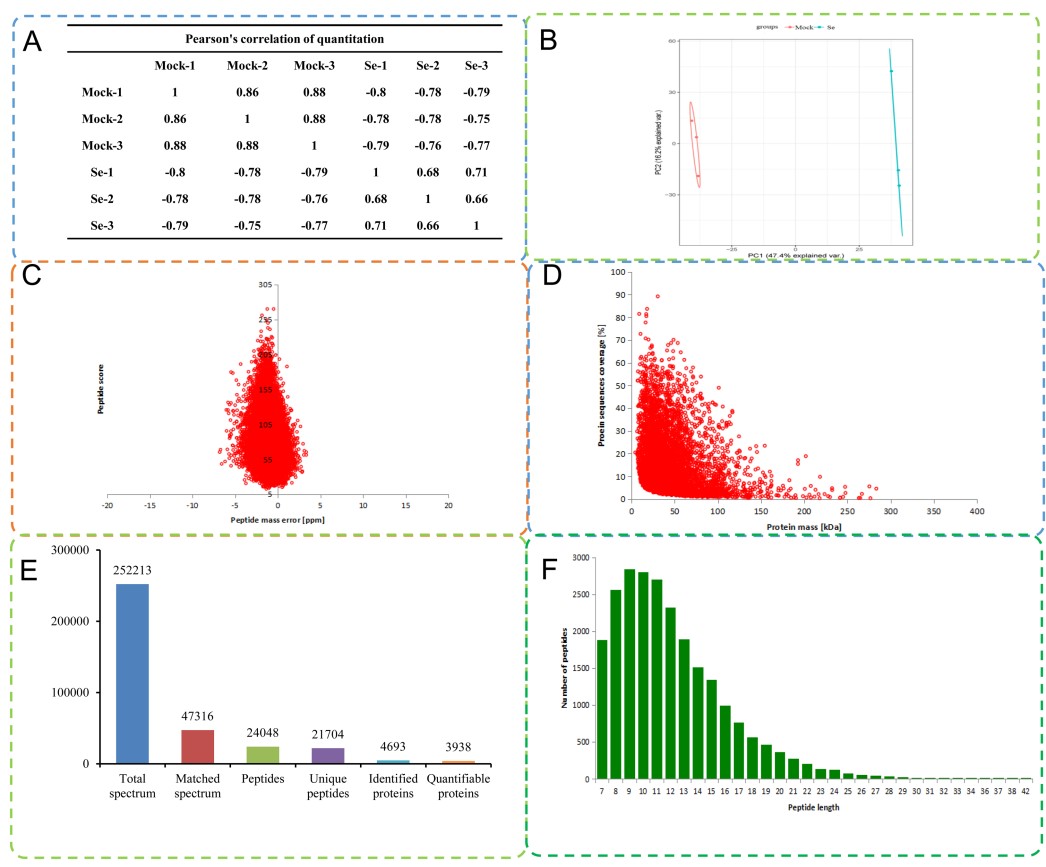

**Figure 1 Quality control (QC) validation of Mass spectrometer (MS) data.** (A) Heatmap of Pearson correlation coefficients from all quantified proteins between each pair of samples. Protein were extracted in three biological replicates for each sample group. All protein samples were trypsin digested and analyzed by HPLC-MS/MS. (B) Two-dimensional scatter plot of PCA (principal component analysis) distribution of all samples using quantified proteins. (C) Relationship between molecular weight and coverage of proteins identified by mass spectrometry. (D) Mass error distribution of all identified peptides. (E) Basic statistical data of MS results. (F) Length distribution of all identified phosphorylated peptides.               

standard to calculate relative fold-differences based on comparative cycle threshold $(2^{-\Delta\Delta Ct})$ values.

## RESULTS

### Primary MS data and quantitative proteome analysis

A total of 252,213 secondary spectra were obtained by MS analysis of the mock treated and Se treated pepper seedlings. A Pearson correlation coefficient analysis indicated high replicability of the experiment (Fig. 1A). A search of the proteome *C. annuum* database indicated 47,316 spectra were available, and the utilization rate of the spectra was 18.8%. A total of 24,048 peptide fragments were identified by spectral analysis, including 21,704 unique peptide fragments (Fig. 1B). Most of the peptides contained 7–20 amino acids, results consistent with the trypsin enzymatic hydrolysis and high-energy collision dissociation fragmentation (Fig. 1C). The molecular weights of the proteins were

negatively correlated with their coverage (Fig. 1D). The first-order mass error of most spectra was less than 10 ppm, in agreement with the high accuracy of orbital well MS. The results indicated high mass accuracy of the MS data (Fig. 1E). Principal component analysis of the quantitative protein data for all samples is presented in Fig. 1F. Detailed information on the identified peptides, including amino acid sequences, protein descriptions, carried charge of peptide, The maximal posterior error probability for peptides (PEP) is listed in Table S2.

A total of 4,693 proteins were identified, among which 3,938 were quantifiable. To understand the functions and characteristics of the proteins identified and quantified in the data, we performed detailed annotation analysis on the basis of GO, protein domain, KEGG pathway, KOG functional classification (eukaryotic orthologous groups), and subcellular localization (Table S3).

## Effects of Se treatment on the global proteome of pepper seedlings

A total of 200 DEPs were identified with a fold-difference expression threshold of 1.5 (Se/Mock ratio $\geq 1.5$ or $\leq 0.667$) and a $t$-test $P$-value < 0.05 (Table S4). All identified proteins and DEPs under Se treatment were grouped into three GO categories (biological process, cellular component, and molecular function) (Fig. 2A). In the biological process category, 1,594 identified proteins and 67 DEPs were involved in "metabolic processes," 1,423 identified proteins and 46 DEPs were involved in "cellular processes," and 1,107 identified proteins and 46 DEPs were involved in "single-organism processes." In the molecular function category, 1,959 identified proteins and 74 DEPs had "catalytic activities," 1,899 identified proteins and 74 DEPs had "binding activities," and 178 identified proteins and one DEP had "structural activities." In the cellular components category, 737 identified proteins and 10 DEPs were "cell"-related proteins; 418 identified proteins and three DEPs were "macromolecular complex"-related proteins; and 402 identified proteins and four DEPs were "organelle"-related proteins. The distribution of the GO annotations of the up-regulated and down-regulated DEPs is shown in Fig. S1. We also used WoLFPSORT (http://wolfpsort.org/) software to determine the subcellular location prediction and classification statistics for all identified proteins and DEPs (Fig. 2B), which were grouped according to their subcellular localizations. All identified proteins were classified into 15 subcellular components, including 1,728 chloroplast-localized proteins, 1,283 cytoplasm-localized proteins, and 784 nuclear-localized proteins. For the DEPs, 13 subcellular components were identified, including 72 cytoplasm-localized DEPs, 44 chloroplast-localized DEPs, and 42 nuclear-localized DEPs.

The expression profiles of the DEPs in six samples are presented in a heat map (Fig. 3A). To reveal the changing trends among the six samples, we assigned all DEPs to one of three clusters (I–III) by using MeV (https://sourceforge.net/projects/mev-tm4/) with the K-means method (Fig. 3B). The proteins in clusters I and II showed up-regulation, whereas the proteins in cluster III were down-regulated in the Se stress treatment replicates. Among the DEPs, 172 were up-regulated and 28 were down-regulated (Figs. 3C and 3D). The 16.6 kDa heat shock protein (HSP) (A0A1U8FR15) and two 18.5 kDa class I HSPs (A0A1U8DWR1, A0A1U8E6M6) were up-regulated over ninefold by Se treatment
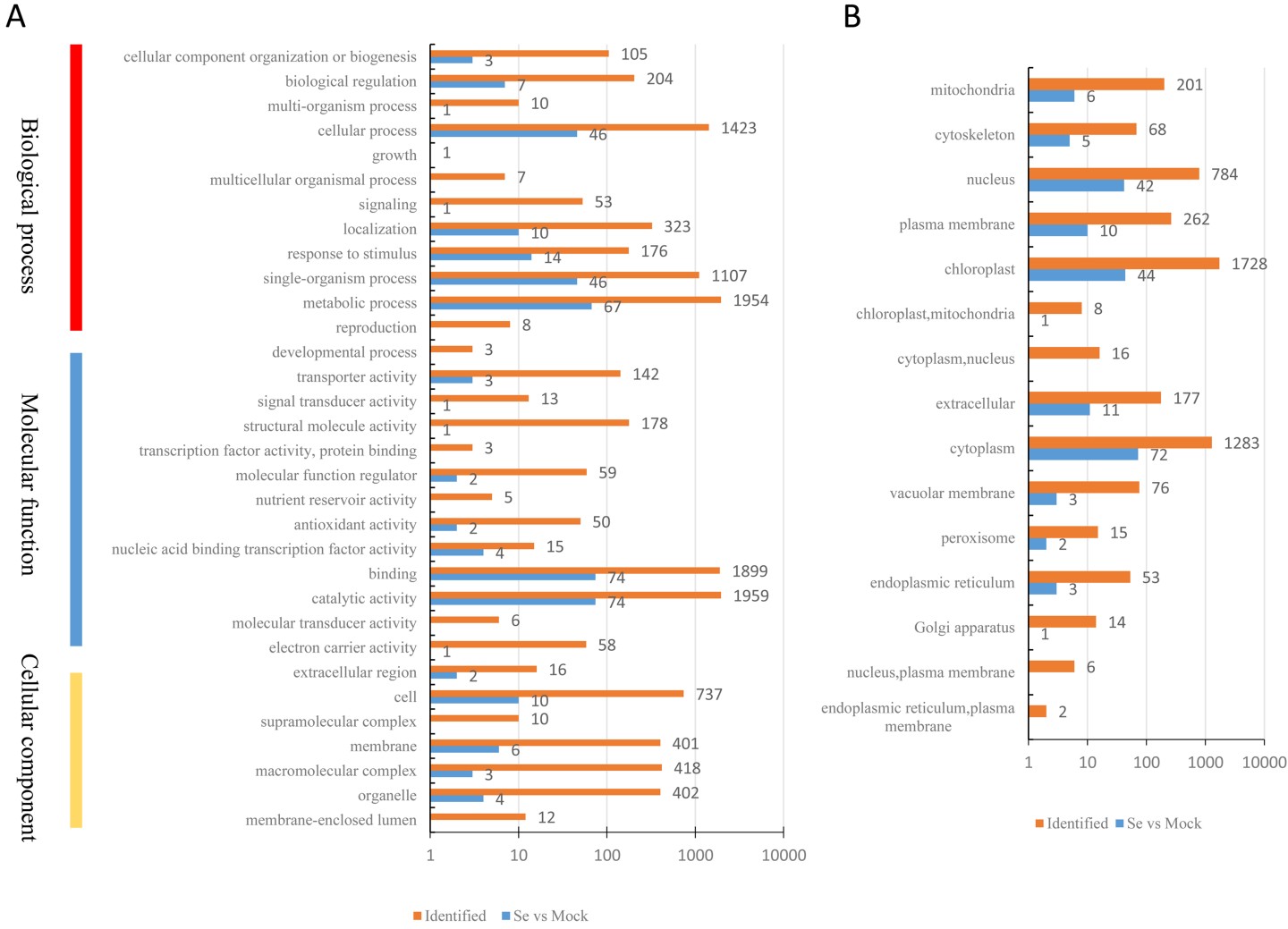

**Figure 2 Classification of all identified proteins and DEPs.** (A) GO analysis of all identified proteins and DEPs. All proteins were classified by GO terms based on three categories: molecular function, biological process and cellular component. (B) Subcellular classify of all identified proteins and DEPs.

compared with the mock treatment. In addition, a glycine-rich protein (A0A2G2ZTC5) and histone H1 protein were down-regulated more than twofold by Se treatment compared with the mock treatment. A total of 136 DEPs were classified into 20 KOG terms. "Post-translational modification, protein turnover, chaperones" contained the largest DEPs (Fig. 4).

## Enrichment analysis of DEPs under Se treatment

To determine whether the DEPs were significantly enriched in some functional types, we performed an enrichment analysis of DEPs by using GO classification, KEGG pathways, and protein domains. Among the DEPs, most up-regulated proteins were enriched in "sequence-specific DNA binding," "iron ion binding," "prephenate dehydrogenase ($NAD^+$/$NADP^+$) activity," "nucleic acid binding transcription factor activity," and "apoplast" (Fig. 5A). For the down-regulated proteins, the top five enriched GO terms were
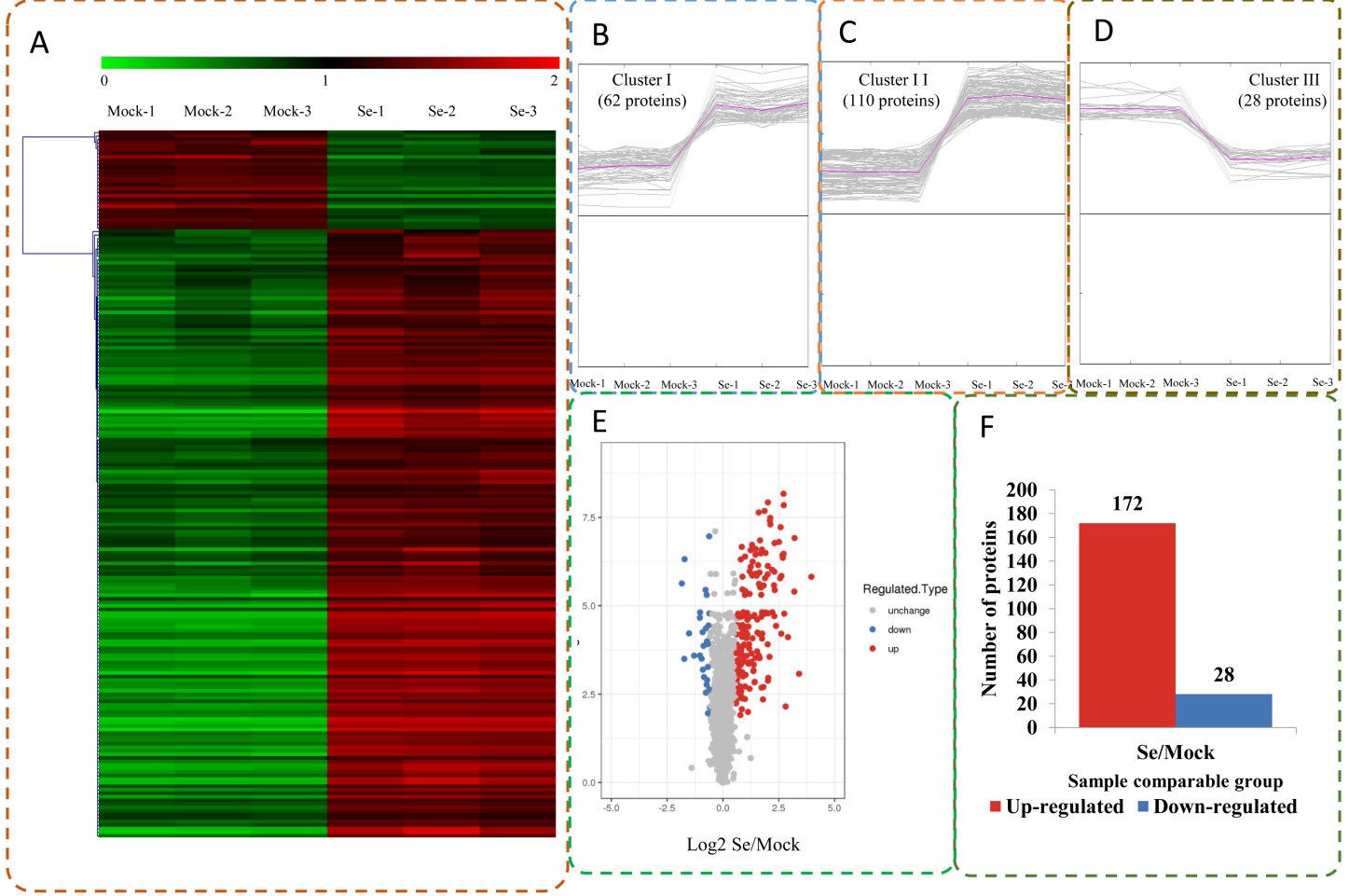

**Figure 3 Impacts Se stress treatment on proteome levels in pepper.** (A) Expression profiles of the DEPs response to Se stress. (B–D) All DEPs were analyzed and clustered into three major Clusters by K-means method. (E) Volcano plot of DEPs. (F) The numbers of up- and down-regulated proteins in the Se treatment seedlings compared to the mock seedlings.

"thiamine-containing compound metabolic process," "chlorophyll metabolic process," "porphyrin-containing compound biosynthetic process," "water-soluble vitamin biosynthetic process," and "vitamin biosynthetic process" (Fig. 5B).

Under Se treatment, 131 DEPs were grouped into different KEGG pathways, seven of which were enriched ($P < 0.05$). For the up-regulated proteins, the DEPs were associated with "protein processing in endoplasmic reticulum," "endocytosis," "sesquiterpenoid and triterpenoid biosynthesis," "SNARE interactions in vesicular transport," and "plant-pathogen interaction" (Fig. 6A). For the down-regulated proteins, the DEPs were associated with "thiamine metabolism" and "porphyrin and chlorophyll metabolism" (Fig. 6B). In the present study, 32 DEPs in the Se treated pepper were identified to be involved in nine metabolic pathways, most of which were significantly up-regulated by Se treatment. However, in the "thiamine metabolism" pathway, four proteins (A0A2G3ALC4, A0A1U8FC73, A0A2G3A131, and A0A2G2Z8I0) were significantly down-regulated by Se treatment (Table 1).

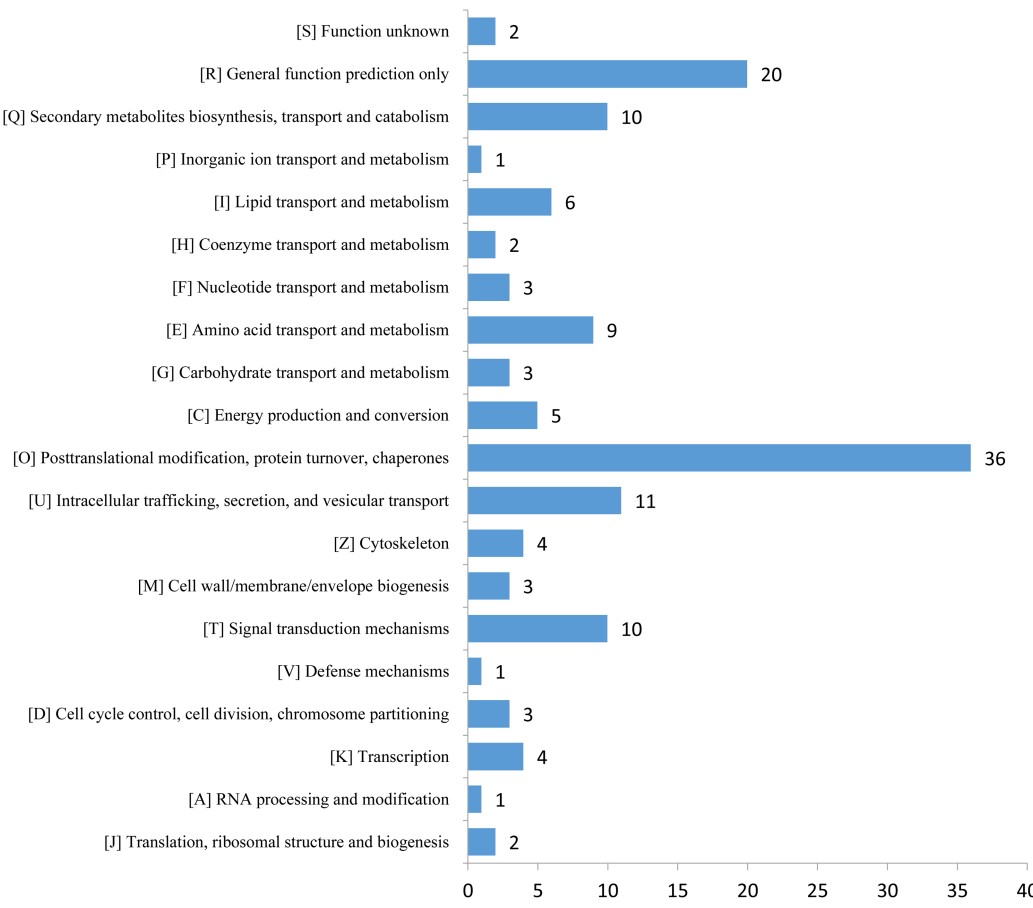

**Figure 4 KOG functional classification chart of differentially expressed proteins.**

Protein domain enrichment analysis revealed that 25 protein domains were enriched in the DEPs (Fig. 7A). The five most enriched protein domains were "alpha crystallin/Hsp20 domain" (13 proteins), "HSP20-like chaperone" (14 proteins), "target SNARE coiled-coil homology domain" (four proteins), "HSP 70 kDa" (five proteins), and "DnaJ domain" (six proteins). Many HSPs were identified according to the results of the protein domain enrichment analysis. In total, 23 HSPs were significantly up-regulated (Fig. 7B). The expression levels of some HSP genes were basically consistent with the proteomic analyses (Fig. S2).

## DISCUSSION

With the increased accumulation of Se in the soil as a result of anthropogenic activity, Se, which has toxic effects on plants at certain concentrations, has gradually become a potential environmental risk factor. In studies of Se, quantitative proteomics analyzed through MS have been applied primarily in analysis of bio-transformation of Se-containing compounds in experimental models such as animals, yeast, and cancer cells (*Zhang et al., 2010*; *Sinha et al., 2016*). In general, proteomic analysis technologies have rarely been applied in the study of Se metabolism in plants. In the present study,

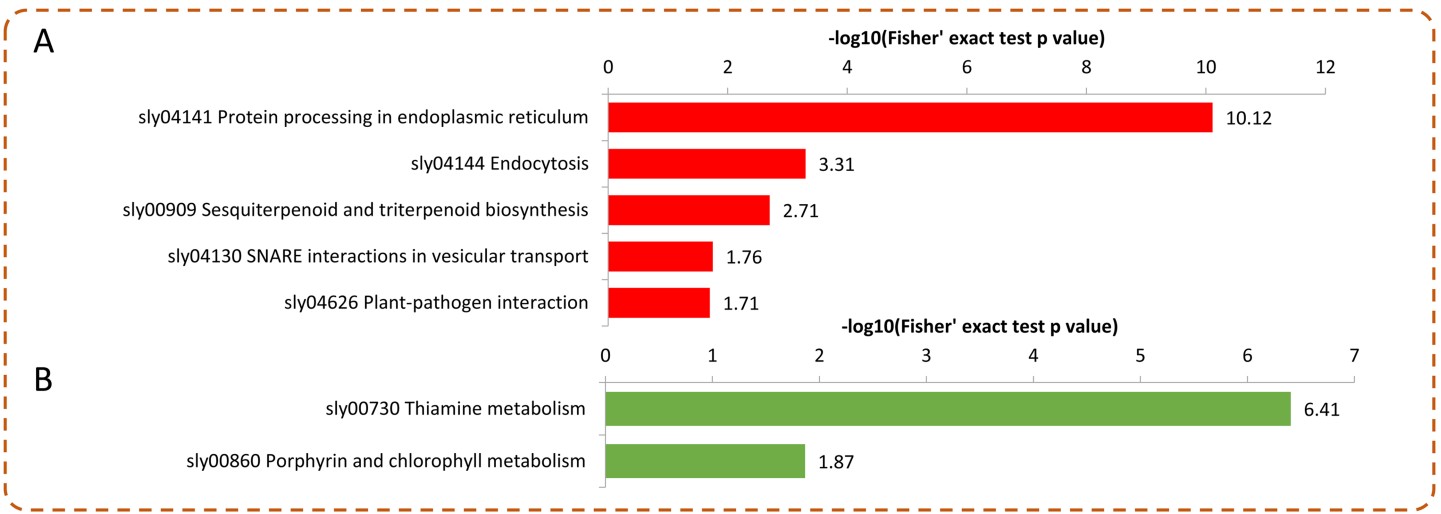

**Figure 5 GO enrichment analysis of DEPs.** Distribution of the up-regulated (A) and down-regulated (B) proteins with GO enrichment analysis.

**Figure 6 KEGG enrichment analysis of the DEPs in pepper after Se stress treatment.** (A) Significantly enriched KEGG terms of the up-regulated proteins. (B) Significantly enriched KEGG terms of the down-regulated proteins.

**Table 1 Identification of the DEPs involved in metabolic pathways.**

| Protein accession | Protein description | Ratio | P-value | MW (kDa) |
|---|---|---|---|---|
| Butanoate metabolism | | | | |
| A0A1U8E166 | "Hydroxymethylglutaryl-CoA lyase, mitochondrial | 1.788 | 0.00039971 | 45.332 |
| A0A2G2YVX0 | Glutamate decarboxylase | 4.962 | 1.6822E-07 | 59.504 |
| Cysteine and methionine metabolism | | | | |
| A0A2G3AEL6 | L-lactate dehydrogenase | 4.042 | 0.00132081 | 37.54 |
| A0A2G3ADN2 | 1-aminocyclopropane-1-carboxylate synthase | 3.506 | 0.00199508 | 54.867 |
| A0A1U8FJ05 | 1-aminocyclopropane-1-carboxylate oxidase 4 | 1.601 | 1.67031E-05 | 36.33 |
| A0A1U8FEU9 | 1-aminocyclopropane-1-carboxylate oxidase 1 | 3.69 | 1.6569E-06 | 36.059 |
| A0A1U8EYM1 | Tyrosine aminotransferase | 2.399 | 0.00038372 | 47.177 |
| A0A1U8E953 | Arginine decarboxylase | 3.279 | 4.8968E-06 | 78.211 |
| A0A1U8FBD0 | Proline dehydrogenase | 2.603 | 0.00046451 | 55.32 |
| A0A2G2ZVP6 | Lipoxygenase | 2.359 | 2.6782E-07 | 102.6 |
| A0A2G2YXE3 | Glyoxysomal fatty acid beta-oxidation multifunctional protein MFP-a | 4.247 | 0.00027878 | 101.81 |
| Glycerophospholipid metabolism | | | | |
| A0A2G2YY43 | Glycerol-3-phosphate 2-O-acyltransferase 4 | 0.625 | 0.00053958 | 51.031 |
| A0A1U8G1E3 | Glycerophosphodiester phosphodiesterase GDPD2 | 2.767 | 2.4846E-07 | 42.766 |
| A0A1U8H0F1 | Glycerol-3-phosphate acyltransferase 5 | 2.083 | 0.00026256 | 55.181 |
| A0A2G2ZQ18 | Putative choline kinase 1 | 2.506 | 0.000083157 | 40.256 |
| Linoleic acid metabolism | | | | |
| A0A2G2ZVP6 | Lipoxygenase | 2.359 | 2.6782E-07 | 102.6 |
| A0A2G2ZBY6 | Lipoxygenase | 3.646 | 2.0736E-08 | 97.904 |
| Phenylpropanoid biosynthesis | | | | |
| A0A2G2YQ27 | Phenylalanine ammonia-lyase | 4.291 | 3.8524E-08 | 78.308 |
| A0A1U8DW23 | Peroxidase | 1.569 | 0.000022029 | 36.129 |
| A0A2G2YUF1 | Retinal dehydrogenase 1 | 2.003 | 3.8333E-06 | 54.692 |
| A0A2G3A835 | Caffeoyl-CoA O-methyltransferase 1 | 1.769 | 0.000020361 | 27.232 |
| Phenylalanine, tyrosine and tryptophan biosynthesis | | | | |
| A0A1U8FBU3 | Arogenate dehydrogenase 1, chloroplastic | 1.76 | 0.0026381 | 42.449 |
| A0A1U8EYM1 | Tyrosine aminotransferase | 2.399 | 0.00038372 | 47.177 |
| A0A2G2YRI9 | Arogenate dehydrogenase 1 | 2.856 | 1.4056E-06 | 45.635 |
| Ubiquinone and other terpenoid-quinone biosynthesis | | | | |
| A0A1U8FWD5 | Putative NAD(P)H dehydrogenase (Quinone) FQR1-like 1 | 1.719 | 0.0124045 | 21.674 |
| A0A1U8EYM1 | Tyrosine aminotransferase | 2.399 | 0.00038372 | 47.177 |
| Sesquiterpenoid and triterpenoid biosynthesis | | | | |
| A0A1U8HFR8 | Vetispiradiene synthase 1 | 6.712 | 1.67142E-05 | 64.165 |
| A0A1U8EWI6 | Uncharacterized protein | 3.689 | 1.67363E-05 | 56.854 |
| Thiamine metabolism | | | | |
| A0A2G3ALC4 | 1-deoxy-D-xylulose-5-phosphate synthase, chloroplastic | 0.607 | 0.000102066 | 76.896 |
| A0A1U8FC73 | Thiamine thiazole synthase, chloroplastic | 0.548 | 0.000136529 | 38.071 |
| A0A2G3A131 | Adenylate kinase 4 | 0.604 | 4.9351E-06 | 26.487 |
| A0A2G2Z8I0 | Phosphomethylpyrimidine synthase | 0.612 | 0.00125775 | 70.093 |

Peer J

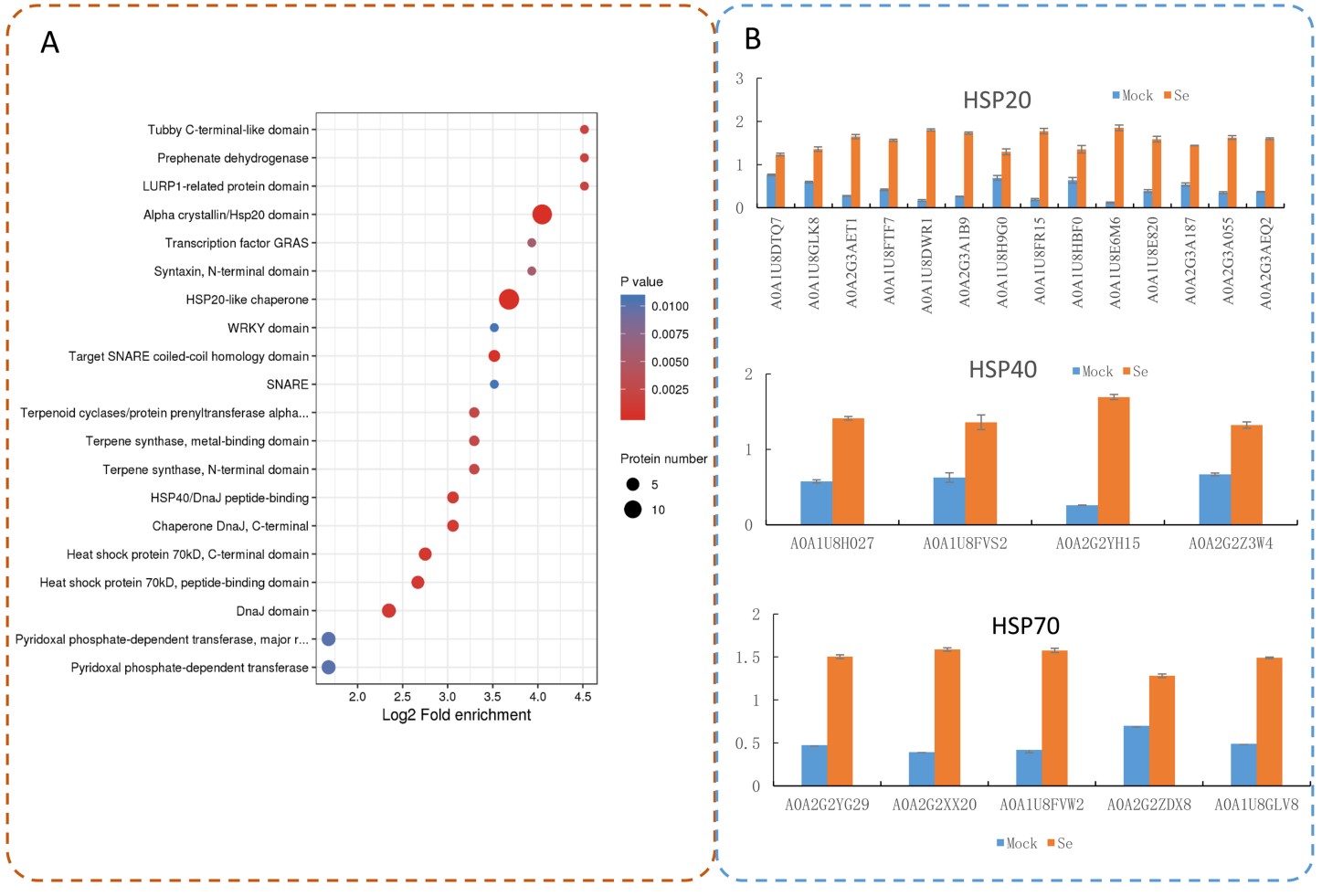

**Figure 7 Domain enrichment analysis of the DEPs in pepper after Se stress treatment.** (A) Protein domain enrichment bubble plot of DEPs. (B) The accumulation of HSP proteins after Se stress treatment.

we conducted a TMT-based quantitative proteomic analysis of the responses of pepper shoots to Se stress. Many DEPs were identified, and a group of proteins potentially involved in Se stress responses were identified. The Se-responsive DEPs and the associated metabolic pathways may play critical roles in Se stress signaling and responses in pepper.

Se is chemically similar to sulfur and is absorbed by plants through similar metabolic pathways (*Van Hoewyk et al., 2008*; *Cakir, Turgut-Kara & Ari, 2016*). Most plants non-specifically absorb Se from the environment through sulfate transporters and assimilate Se into organic forms of Se through the S metabolic pathway (*Cappa et al., 2014*). To date, studies on Se-tolerance mechanisms have focused on some Se-hyperaccumulating plants (*Freeman et al., 2010*; *Sabbagh & Van Hoewyk, 2012*; *Cappa et al., 2015*). In *Cardamine hupingshanensis* seedlings, the expression of the *sulfite oxidase* (*SOX*) gene in the roots is up-regulated after the addition of selenite, thus indicating that selenite may first be converted to selenate, and then the selenate is metabolized (*Zhou et al., 2018*).

Plants have evolved several efficient and complex strategies for dealing with different abiotic stresses, including Se stress (*Chen et al., 2002*; *Lyons, Stangoulis & Graham, 2004*; *Pérez-Clemente et al., 2013*; *Schiavon et al., 2013*). GO analysis revealed that 14 proteins were associated with "response to stimulus," including 11 up- and three down-regulated proteins significantly altered by Se stress in pepper. Among the proteins, a peroxidase (A0A1U8DW23) and a glutathione peroxidase (A0A2G3AE16) were up-regulated over 1.5-fold by Se treatment, thus suggesting that reactive oxygen species accumulate after Se treatment. In the present study, the expression of two pathogenesis-related proteins (STH-21, A0A1U8GEH7; PR4, I6VW44) significantly decreased. Pathogenesis-related proteins are a class of stress-tolerant proteins that are promising tools for plant genetic engineering (*Ali et al., 2018*). KOG analysis revealed that 36 proteins were related to the term "posttranslational modification, protein turnover, chaperones." The identified DEPs may be involved in plant responses to Se stress. In pepper, nine "amino acid transport and metabolism"-related proteins and 10 "secondary metabolites biosynthesis, transport and catabolism"-associated proteins were identified. The changes in the DEPs suggested that Se stress may influence metabolite content in pepper.

Plants are thought to produce specific metabolites in response to Se stress (*Fernandes et al., 2018*; *Pilon-Smits et al., 2009*). Transcriptomic analysis of Se-treated *Arabidopsis thaliana* has revealed that sulfur content decreases while the expression of sulfur absorption and metabolism genes increases, and the signaling pathways for ethylene and jasmonic acid respond to Se stress (*Van Hoewyk et al., 2008*). Small RNA and degradome sequencing analysis of *Astragalus chyrsochlorus* callus has revealed that miR167a, miR319, miR1507a, miR4346, miR7767-3p, miR7800, miR9748, and miR-n93 target transcription factors, disease resistance proteins, cysteine synthase, plant hormone signal transduction, plant-pathogen interaction, and sulfur metabolism pathways in response to Se stimuli (*Cakir, Candar-Cakir & Zhang, 2016*). In the present study, KEGG enrichment analysis revealed that seven pathways, including the sesquiterpenoid and triterpenoid biosynthesis pathway, plant-pathogen interaction, and thiamine metabolism pathway, were enriched after Se treatment. In addition, 32 DEPs involved in nine metabolic pathways were identified. Nine up-regulated DEPs were associated with cysteine and methionine metabolism. The main cause of plant Se poisoning is thought to be the incorrect synthesis of selenomethionine and selenocysteine (SeCys) into proteins, thereby causing changes or instability in the structures of proteins (*Sabbagh & Van Hoewyk, 2012*; *Van Hoewyk, 2013*). High concentrations of SeCys in cells would lead to Se poisoning, and SeCys transformation is a direct means of Se detoxification (*Pilon et al., 2003*; *Tamaoki, Freeman & Pilon-Smits, 2008*; *Sabbagh & Van Hoewyk, 2012*). Terpenoids, the most diverse class of chemicals produced by plants, are involved in protection against various abiotic factors (*Lange, 2015*; *Tholl, 2015*). For example, terpenoids are considered to be important defensive metabolites in *Eucalyptus froggattii* seedlings (*Goodger, Heskes & Woodrow, 2013*). Four upregulated DEPs were associated with "ubiquinone and other terpenoid-quinone biosynthesis" and "sesquiterpenoid and triterpenoid biosynthesis," thus indicating that Se stress influenced the accumulation of some terpenoids.

Heat shock proteins were initially defined as proteins rapidly up-regulated by heat stress (*Hartl & Hayer-Hartl, 2002*). Studies increasingly show that HSP concentrations in plants increase rapidly as environmental conditions deteriorate (*Murakami et al., 2004*; *Hu, Hu & Han, 2009*; *Lee, Yun & Kwon, 2012*). HSPs are a class of evolutionarily conserved proteins that can be divided into five families according to molecular weight and sequence homology: small HSPs (molecular weight from 15 to 42 kDa), HSP60, HSP70, HSP90, and HSP100 (*Boston, Viitanen & Vierling, 1996*; *Wang et al., 2004*; *Waters, 2013*). HSP90 and HSP70 are essential for plant resistance to pathogen infections (*Kanzaki et al., 2003*; *Noël et al., 2007*; *Chen et al., 2010*). AtHSP17.6A is induced by osmotic stress, and PtHSP17.8 is involved in tolerance to heat and salt stress (*Sun et al., 2001*; *Li et al., 2016*). Among the Se-stress-induced DEPs in pepper, several HSPs were identified. A total of 14 HSP20, four HSP40, and five HSP70 proteins were significantly upregulated by salt stress, thus suggesting molecular cross-talk between heat shock responses and Se stress.

## CONCLUSIONS

A TMT-based proteomic method was used to investigate changes in protein levels between control and Se treated pepper seedlings. In total, 4,693 proteins and 200 DEPs were identified. A number of DEPs were found to be mainly involved in responses to stress and metabolic processes. Our results provide basic tools for identifying candidate proteins and the molecular mechanisms of the Se stress response in pepper plants.

## ACKNOWLEDGEMENTS

We are grateful to the PTM Biolabs company for technical support. We thank International Science Editing for editing this manuscript.

### Funding

This work was funded by the Natural Science Foundation of Zhejiang Province, China, Grant No. LQ17C150003. The National Natural Science Foundation of China, Grant No. 31701967, the key research and development program of Zhejiang Province, China, Grant No. 2017C02018 and the key research and development program of Hangzhou, China, Grant No. 20180416A07. The funders had no role in study design, data collection and analysis, decision to publish, or preparation of the manuscript.

### Grant Disclosures

The following grant information was disclosed by the authors:
Natural Science Foundation of Zhejiang Province: LQ17C150003.
National Natural Science Foundation of China: 31701967.
Key research and development program of Zhejiang Province, China: 2017C02018.
Key research and development program of Hangzhou, China: 20180416A07.

### Competing Interests

The authors declare that they have no competing interests.

## Author Contributions

- Chenghao Zhang conceived and designed the experiments, analyzed the data, contributed reagents/materials/analysis tools, prepared figures and/or tables, approved the final draft.
- Baoyu Xu performed the experiments, contributed reagents/materials/analysis tools, prepared figures and/or tables, approved the final draft.
- Wei Geng performed the experiments, contributed reagents/materials/analysis tools, prepared figures and/or tables, approved the final draft.
- Yunde Shen performed the experiments, contributed reagents/materials/analysis tools, prepared figures and/or tables, approved the final draft.
- Dongji Xuan analyzed the data, authored or reviewed drafts of the paper, approved the final draft.
- Qixian Lai analyzed the data, authored or reviewed drafts of the paper, approved the final draft.
- Chenjia Shen analyzed the data, authored or reviewed drafts of the paper, approved the final draft.
- Chengwu Jin conceived and designed the experiments, analyzed the data, contributed reagents/materials/analysis tools, prepared figures and/or tables, approved the final draft.
- Chenliang Yu conceived and designed the experiments, contributed reagents/materials/analysis tools, prepared figures and/or tables, approved the final draft.

## Data Availability

The raw data is available at ProteomeXchange: PXD013257.

## Supplemental Information

Supplemental information for this article can be found online at http://dx.doi.org/10.7717/peerj.8020#supplemental-information.

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
