# Peer review of "Comparative proteomic analysis of pepper (Capsicum annuum L.) seedlings under selenium stress"

_PeerJ, doi:10.7717/peerj.8020_

## Round 0.1 · original submission · Minor Revisions

Thank you for your submission to PeerJ. Although all 3 reviews are bried, as you can see, they collectively make suggestions for revision. Please attend to all their comments in a full rebuttal and edited revision.

Reviewer 1 ·

Basic reporting

it is satisfactory.

Experimental design

it is satisfactory.

Validity of the findings

Real-Time PCR (qPCR) Verification is neede.Some differential proteins should be selected to perform the mRNA expression level verification in order to validate the labeling results.

Additional comments

No comments.

Reviewer 2 ·

Basic reporting

The manuscript entitled "Comparative Proteomic Analysis of Pepper (Capsicum
annuum L.) Seedlings under Selenium Stress" by Chenghao et al. studies the regulated protein expression in Selenium stress in pepper plant. Overall the manuscript is satisfactory and well referenced & relevant to the concept.
However, I had some minor comments which I mentioned in the next section, after that the manuscript is acceptable for publication.

Experimental design

Satisfactory.

Validity of the findings

Satisfactory.

Additional comments

In table 1, what is the cut-off ratio set up by author to include the peptide candidate in the table?

Did author got any percent coverage (% coverage) of peptide and can highlight the peptide sequences in full length protein sequence that got in Mass spec for the DEP that are highly regulated say 2 times the control.

Did author see any change in band pattern of total protein run on SDS-PAGE and probed by silver staining?

Background and borders of figures are unusual. Please keep it in canonical white background.

Reviewer 3 ·

Basic reporting

The manuscript is well written. There are no grammatical errors. However, the authors should carefully peruse the text to ensure that all Latin/scientific names are italicized.

Please check all the figure legends. the ones for Figure 2 and 3 are incomplete.

Also the text in the figures is unreadable unless zoomed in considerably. If possible please increase the resolution of the figures.

Experimental design

Experimental design is well documented.

Validity of the findings

The authors discuss their findings in great detail. However, the discussion is limited as each type of DEP is discussed separately. The manuscript will have more impact if the authors could include a more global view of the processes in the plant at the end of their discussion.

Additional comments

This manuscript is very well written and shows showcases a well organized study into stress response in plants. I will be very interested to see further studies based on your current endeavor in the pepper plant and comparisons with other plants undergoing similar stress.

---

## Round 0.2 · Minor Revisions

Dear Dr. Yu,

It is my pleasure to inform you that your manuscript titled "Comparative proteomic analysis of pepper (Capsicum annuum L.) seedlings under selenium stress" is largely Acceptable, pending the Section Editor requests below.

My best regards,
Dr. Gilda Eslami

Reviewer 1 ·

Basic reporting

Clear and unambiguous

Experimental design

Original primary research within Aims and Scope of the journal.

Validity of the findings

Conclusions are well stated, linked to original research question & limited to supporting results

Reviewer 2 ·

Basic reporting

Manuscript has improved after revision. All questions have been answered satisfactorily by the authors.

Experimental design

Satisfactory

Validity of the findings

Satisfactory

Reviewer 3 ·

Basic reporting

N/A

Experimental design

N/A

Validity of the findings

N/A

Additional comments

The authors have satisfactorily answered the issues raised by me in the last review.

---

## Round 0.3 · accepted · Accept

Dear Dr. Yu,
Hello,

I am pleased to inform you that your manuscript titled "Comparative proteomic analysis of pepper (Capsicum annuum L.) seedlings under selenium stress" has been accepted for publication.

My best regards,
Dr. Gilda Eslami

Reviewer 2 ·

Basic reporting

The manuscript has significantly improved after revision. The authors had answered all the questions.

Experimental design

Valid

Validity of the findings

No Comments

Additional comments

The manuscript has more speculative/predictive datasets of MS identified proteins which is not of much information rather than jargon. However, it may be accepted for publication.

Reviewer 3 ·

Basic reporting

The authors have addressed my concerns from the previous reviews

Experimental design

The authors have addressed my concerns from the previous reviews

Validity of the findings

The authors have addressed my concerns from the previous reviews

Additional comments

The authors have addressed my concerns from the previous reviews